# Initial functional disability as a 1-year prognostic factor in geriatric patients hospitalized with SARS-CoV-2 infection

**Olivier Brière**[1]*, **Marie Otekpo**[1], **Marine Asfar**[1], **Jennifer Gautier**[1],
**Guillaume Sacco**[3,4¤a], **Cédric Annweiler**[1,2,5,6¤b], on behalf of the GERIA-COVID study group[¶]

**1** Department of Geriatric Medicine and Memory Clinic, Research Center on Autonomy and Longevity, University Hospital, Angers, France, **2** UNIV ANGERS, University of Angers, Angers, France, **3** University Côte d'Azur, Nice, France, **4** Department of Geriatric Medecine and Brain Clinic, Nice, France, **5** Gérontopôle Autonomie Longévité des Pays de la Loire, Nantes, France, **6** Department of Medical Biophysics, Schulich School of Medicine and Dentistry, Robarts Research Institute, The University of Western Ontario, London, ON, Canada

¤a Current address: Department of Geriatric Medecine and brain clinic, Nice, France
¤b Current address: Department of Geriatric Medicine and Memory Clinic, Research Center on Autonomy and Longevity, University Hospital, Angers, France
¶ The Complete Membership of the author group can be found in the Acknowledgments section.
* olivier.briere@chu-angers.fr

**Data Availability Statement:** Patient level data are freely available at DRCI@chu-angers.fr to qualifying researchers registered with an appropriate

## Abstract

### Background

SARS-CoV2 infection has affected many older people and has required us to adapt our practices to this new pathology. Initial functional capacity is already considered an important prognostic marker in older patients particularly during infections.

### Aim

The objective of this longitudinal study was to determine whether baseline functional disability was associated with mortality risk after 1 year in older patients hospitalized for COVID-19.

### Methods

All COVID-19 patients admitted to the geriatric acute care unit of Angers University Hospital, France, between March-June 2020 received a group iso-ressource (GIR) assessment upon admission. Disability was defined as a GIR score≤3. All-cause mortality was collected after 1 year of follow-up. Covariables were age, sex, history of malignancies, hypertension, cardiomyopathy, number of acute diseases at baseline, and use of antibiotics or respiratory treatments during COVID-19 acute phase.

### Results

In total, 97 participants (mean±SD 88.0+5.4 years; 49.5% women; 46.4% GIR score≤3) were included. 24 of the 36 patients who did not survive 1 year had a GIR score ≤ 3 (66.7%;

institution and who submit a proposal with a valuable research question. There is no personal identification risk within this anonymized raw data, which is available after notification and authorization of the competent authorities.

**Funding:** The author(s) received no specific funding for this work.

**Competing interests:** The authors have declared that no competing interests exist.

P = 0.003). GIR score≤3 was directly associated with 1-year mortality (fully adjusted HR = 2.27 95% CI: 1.07–4.89). Those with GIR≤3 at baseline had shorter survival time than the others (log-rank P = 0.0029).

## Conclusions

Initial functional disability was associated with poorer survival in hospitalized frail elderly COVID-19 patients.

## Clinical trial registration

ClinicalTrials.gov: NCT04560608 registered on September 23, 2022

## Introduction

In December 2019, a new coronavirus SARS-CoV-2 was discovered in Wuhan (China), and caused millions of deaths worldwide [1]. In April 2020, the SARS-CoV-2 (COVID-19) infection was declared a pandemic emergency by the World Health Organization [2]. The various clinical forms of the infection are usually mild, but more often symptomatic and severe in older polymorbid people [3–5]. Prognostic factors were initially unknown but some, including comorbidities, were quickly identified [6].

During the pandemic, the number of hospital beds available was overwhelmed by the influx of patients with severe COVID-19, forcing to select the profile of patients eligible for hospitalization, i.e. those who would benefit from their hospital stay. An infection such as COVID-19 can lead to a succession of episodes of decompensation of chronic conditions in older patients. The International Classification of Impairments, Disabilities and Handicaps (ICHD) has shown that every organ impairment leads to disability, especially in older people [7]. It is therefore recognized that the initial functional capacity is a potent prognostic marker in older patients, particularly in the case of acute infection [8–10]. The whole question is whether functional disability on hospital admission is associated with life-threatening in SARS-CoV2 infections. Estimating the pre-morbid functional state of older patients could then help to better orient patients in the care pathways most suited to their condition and their hope of survival and recovery in the context of health crisis.

In France, the most commonly used scale for assessing functional disability is the AGGIR grid ('Autonomie Gérontologique Groupe Iso-Ressources'). [7]. It is commonly used in emergency departments and intensive care units to estimate the disability of older patients [8].

We hypothesized that the initial functional disability may serve as a prognostic factor in older patients with COVID-19. Our objective was to determine whether the presence of an initial functional disability according to the AGGIR grid was associated with an increase 1-year mortality in geriatric patients with COVID-19 in the GERIA-COVID cohort (GERIAtric Patients Hospitalized for COVID-19) during the first wave of the pandemic in France.

## Methods

The GERIA-COVID study is a longitudinal observational study conducted in the geriatric acute care unit dedicated to COVID-19 patients in the University Hospital of Angers, France, during the first wave of the COVID-19 pandemic (ClinicalTrials.gov NCT04560608). All baseline characteristics were retrospectively collected from patients' medical records, including data related

to COVID-19, vital measurements, medical history, standardized geriatric assessment, usual treatments, standardized clinical examinations and blood tests on hospital admission.

## Study population

Patients eligible in the GERIA-COVID study were adults aged 75 years and over admitted into the geriatric acute care unit of the University Hospital of Angers, France, during the first wave of the pandemic (between March and June 2020) who were no objection for using anonymized clinical and biological data for research purpose.

The inclusion criteria for the present analysis were as follows: (i) COVID-19 diagnosed with RT-PCR and/or chest-CT scan; (ii) data available on the vital status 1 year after the diagnosis of COVID-19. Ninety-seven patients were diagnosed with COVID-19 during the study period in the unit, and were recruited in the GERIA-COVID study. This analysis included all of them.

## Functional disability

Initial functional disabilities were examined using the AGGIR grid upon hospital admission by paramedical staff of the geriatrics department [9]. It includes ten discriminating variables (ie, coherence, orientation, toilet, dressing, feeding, urinary and fecal continence, communication, displacement) to classify older adults into 6 Iso-Resource Groups (GIR) [7]. Each item is rated on a scale of 3 levels (A: done alone completely, usually, correctly; B: partially done, not usually, not correctly; C: Does not do). An algorithm ultimately classifies patients into one GIR group. GIR 6 corresponds to fully independent patients, and GIR 1 to fully dependent patients. In the present study, functional disability was defined as a GIR score ≤3 out of 6 [10].

## Overall 1-year mortality

Follow-up started from the day of COVID-19 diagnosis for each patient and continued for up to 12 months or until death when applicable. Vital status was recovered by contacting the patients and their relatives by telephone, and by monitoring the National Institute of Statistics and Economic Studies (INSEE) register (https://www.insee.fr/fr/information/4190491).

## Covariates

Potentials confounders were age, sex, number of acute health issues on hospital admission, history of hypertension, cardiomyopathy, malignancies, and use of antibiotics and respiratory treatments during COVID-19. The number of acute health issues on admission was also recorded (whatever their nature or site). History of cardiomyopathy, malignancies and hypertension were recorded on admission from patient medical file and caregiver interviews. Use of antibiotics and pharmacological treatments of respiratory disorders was systematically recorded from hospital prescriptions.

## Statistical analysis

The participants' characteristics were summarized using means and standard deviations (SD) or frequencies and percentages, as appropriate. Firstly, comparisons between participants separated according to baseline GIR at the end of 1st year of the diagnosis of COVID-19 were performed using Chi-square test (or Fisher exact test) or Student t test (or Mann-Whitney Wilcoxon test according to the normality assessment), as appropriate. Secondly, adjusted Cox regressions were used to examine the associations of baseline GIR (independent variable) with 1-year mortality (dependent variable). The models produce a survival function that provides the probability of death at a given time for the characteristics supplied for the independent

variables. Finally, the elapsed time to death was studied by survival curves computed according to Kaplan-Meier method and compared by log-rank test. P-values<0.05 were considered significant. All statistics were performed using SAS® version 9.4 software (Sas Institute Inc) and R (R core Team, 2018).

### Ethics

The study was conducted in accordance with the ethical standards set forth in the Helsinki Declaration (1983). No participants or relatives objected to the use of anonymized clinical and biological data for research purposes. Ethics approval was obtained from the Ethics Board of the University Hospital of Angers, France (2020/100). The study protocol was also declared to the National Commission for Information Technology and civil Liberties (CNIL; ar 20-0087v0).

## Results

Ninety-seven patients were consecutively diagnosed with COVID-19 in our unit during the study period and they were all enrolled in the GERIACOVID cohort study. All met the eligibility criteria and were included in the present analysis (mean±standard deviation, 88.0±5.4 years, 49.5% women, 46.4% with GIR score≤3). The vital status of all participants was known after one year. Among them, 61 participants (62.8%) survived COVID-19 after one year of follow-up while 36 participants died (37.1%).

Table 1 shows the characteristics of the participants according to their vital status after one year of follow-up. One-year mortality was higher in the group with disability (GIR score≤3) on admission for SARS-COV2 infection (53.3% versus 23.1% respectively, p = 0.002). Patients who died at 1 year had also more often a history of malignancies (p = 0.011).

Table 2 shows a direct association between disability (GIR score≤3) baseline and 1-year mortality. The hazard ratio (HR) for mortality was 2.69 [95% confidence interval (CI): 1.34;15.38] (P = 0.005) in the unadjusted model, and 2.27 [95%CI: 1.07;4.89] (P = 0.033) after adjustment for all potential confounders. The history of malignancies was also associated with greater mortality risk (HR = 2.47, P = 0.011) in the adjusted model.

**Table 1. Characteristics and comparison of COVID-19 patients (n = 97) separated into two groups according to a GIR score ≤ 3.**

|  | Total cohort (n = 97) | Disability (GIR score ≤ 3) | | P-value* |
|---|---|---|---|---|
|  |  | No (n = 52) | Yes (n = 45) |  |
| **Demographical data** |  |  |  |  |
| Age (years), mean±SD | 88.0 ± 5.4 | 87.0 ± 5.5 | 89.1 ± 5.2 | 0.061 |
| Female gender | 48 (49.5) | 24(46.2) | 24(53.3) | 0.481 |
| **Comorbidities** |  |  |  |  |
| Hematological and solid cancers | 33 (34.0) | 14(26.9) | 19(42.2) | 0.113 |
| Hypertension | 62 (63.9) | 33(63.5) | 29(64.4) | 0.920 |
| Cardiomyopathy | 52 (53.6) | 28(53.9) | 24(53.3) | 0.960 |
| **Hospitalization** |  |  |  |  |
| Number of acute health issues at hospital admission, med[IQ] | 3 [2–4] | 2 [1.5–4] | 3 [2–4] | 0.410 |
| Use of antibiotics† | 65 (67.0) | 31(59.6) | 34(75.6) | 0.096 |
| Use of pharmacological treatments of respiratory disorders‡ | 11 (11.3) | 5(9.6) | 6(13.3) | 0.565 |
| **1-year mortality** | 36(37.1) | 12(23.1) | 24(53.3) | **0.002** |

Data presented as n (%) where applicable; COVID-19: Coronavirus Disease 2019; GIR: Iso Resource Groups; *: between-group comparisons based on Chi-square test (or Fisher exact test) or Student *t* test (or Mann-Whitney Wilcoxon test according to the normality assessment), as appropriate; †: quinolones, beta-lactams, sulfonamides, macrolides, lincosamides, aminoglycosides, among others; ‡: beta2-adrenergic agonists, inhaled corticosteroids, antihistamines, among others.

**Table 2. Multiple Cox proportional-hazards model showing the hazard ratio for 1-year mortality (dependent variable) according to baseline GIR (independent variable), adjusted for participants' characteristics (n = 97).**

| | 1-year mortality | | | | | |
| --- | --- | --- | --- | --- | --- | --- |
| | Unadjusted model | | Fully-adjusted model | | Model with backward selection method | |
| | HR [95% CI] | P-value | HR [95% CI] | P-value | HR [95% CI] | P-value |
| GIR score ≤ 3 | 2.69 [1.34;5.38] | **0.005** | 2.27 [1.07;4.89] | **0.033** | 2.54 [1.26; 5.11] | **0.009** |
| Age | 1.04 [0.98;1.11] | 0.226 | 1.05 [0.98;1.13] | 0.186 | | |
| Female gender | 0.67 [0.34;1.30] | 0.232 | 0.52 [0.25;1.11] | 0.092 | | |
| History of cancer | 2.59 [1.34;4.98] | **0.005** | 2.47 [1.23;4.96] | **0.011** | 2.37 [1.22; 4.58] | **0.011** |
| History of hypertension | 1.13 [0.56;2.26] | 0.734 | 1.26 [0.60;2.63] | 0.545 | | |
| History of cardiomyopathy | 1.11 [0.58;2.15] | 0.748 | 1.00 [0.50;2.00] | 0.998 | | |
| Number of acute health issues at hospital admission | 1.22 [0.99;1.49] | 0.060 | 1.18 [0.94;1.48] | 0.154 | | |
| Use of antibiotics* | 1.67 [0.79;3.55] | 0.183 | 1.51 [0.65;3.51] | 0.340 | | |
| Use of pharmacological treatments of respiratory disorders† | 1.46 [0.57;3.76] | 0.431 | 1.76 [0.64;4.83] | 0.271 | | |

CI: confidence interval; COVID-19: coronavirus disease 2019; GIR: Iso Resource Groups; HR: hazard ratio; *: quinolones, beta-lactams, sulfonamides, macrolides, lincosamides, aminoglycosides, among others; †: beta2-adrenergic agonists, inhaled corticosteroids, antihistamines, among others.

Finally, the Kaplan-Meier distributions (Fig 1) showed that patients with COVID-19 with GIR score≤3 at baseline had shorter survival times than the others [log-rank p = 0.0029].

## Discussion

The main result of this longitudinal study is that, irrespective of all measured potential confounders, disability (defined as a GIR score ≤ 3) in early COVID-19 was associated with 1-year mortality among geriatric patients.

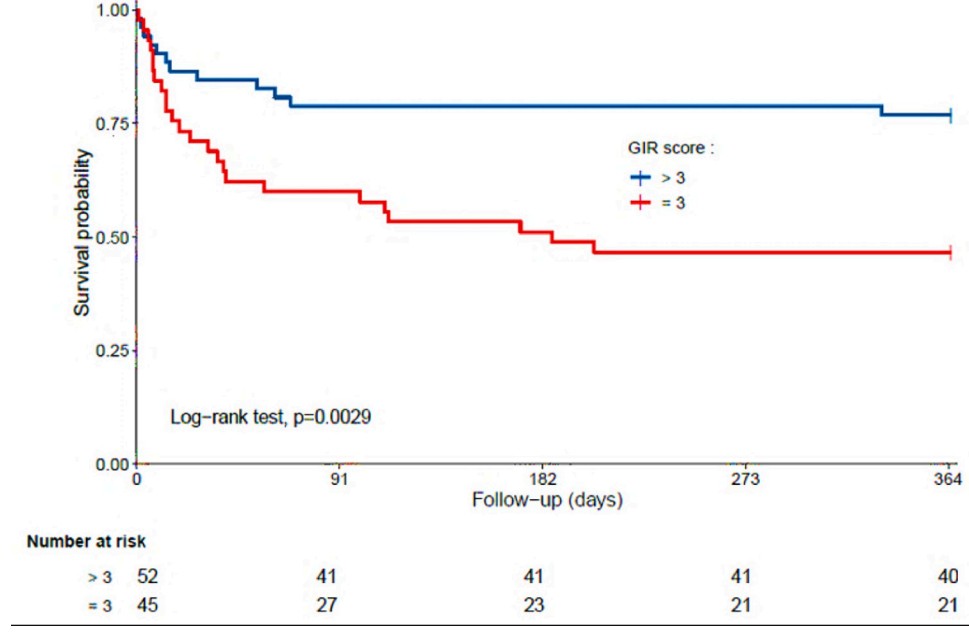

**Fig 1. Kaplan-Meier estimates of the cumulative probability of COVID-19 participants' survival according to GIR (n = 97).**

The association between mortality and disability during acute diseases has been extensively investigated, mainly in intensive care unit (ICU), and is well-recognized to complement the effect of age to predict mortality in older adults following an acute disease [11]. In geriatric patients > 80 years admitted to ICU, the 1-month mortality was associated with frailty, and functional disabilities [12]. In another study, predictive factors of 1-year mortality in 420 patients > 70 years admitted to ICU were mainly related to abilities in daily living, including Katz ADL score, Barthel index and GIR score [13]. However these previous studies were limited by a selection bias as highly dependent patients are generally not admitted to ICU. Another study, focusing on all hospitalized elderly patients > 65 years affected by pneumonia found a direct association between functional disability (with lower Barthel Index) and higher mortality risk [14]. COVID-19 has sparked scientific excitement with many articles highlighting various prognostic factors in older adults [15] such as the female gender, qSOFA score > 2, specific COVID-19 changes on chest CT-scan [16], use of VKA treatment [17], hypercalcemia [18], early inflammatory syndrome with elevated CRP levels [19] but, curiously, very few studies have investigated the possible role of initial functional disability. The qSOFA is a score based on 3 criteria of respiratory, hemodynamic and neurological function, to identify patients at high risk of hospital mortality from infection [20]. Zerah et al. reported, in a cohort of 821 older patients hospitalized for over 70 years with confirmed COVID-19, an increase in hospital mortality due to functional impairment (ADL < 4) [16]. In another large study of 2359 older patients > 70 years admitted into ICU for COVID-19, the group with both pre-existing frailty and pre-existing disability (ADL score < 6) exhibited extremely high 3-month mortality risk [21]. Thus, these few previous results were consistent with our present analysis focused on a geriatric population aged ≥75 years.

Disability is defined as the loss of functional and cognitive capacities, ie the possibility and ability to perform daily activities in an appropriate way, and requiring assistance to carry out acts of daily living [22]. Disability affects more than 50% of patients over age 65 [23], and increases the risks of mortality and morbidity [23, 24]. It is generally the adverse consequence of both physiological and pathological aging. For Bouchon et al., aging is an unstable balance, with a vital functional reserve that decreases with the evolution of chronic diseases and after each acute decompensation [25]. It appears that the high and uncontrolled release of pro-inflammatory cytokines (such as IL-6, IL-1, TNF-$\alpha$) in COVID-19, ie the cytokine storm, is one of the main mechanisms explaining the severity of the disease in elderly patients [26], causing acute respiratory distress syndrome and death. Advancing in age is usually already associated with increased levels of pro-inflammatory cytokines, and reduced levels of systematic anti-inflammatory cytokines, creating a chronic state of inflammation called 'inflamm-aging' [27]. It is likely that the dysregulation of the cytokine homeostasis in 'inflamm-aging" may play a critical role in the risk of cytokine storm, and therefore in the risk of reduced functional reserve according to the Bouchon model. Maintaining functional capacity is an important indicator of elderly health.

We also found that the history of malignancies predicted greater 12-month mortality in older adults with COVID-19. This result is fully consistent with previous literature that retained this variable as on the main prognostic factor in COVID-19 patients, particularly during the first wave of the pandemic [13, 28–30].

Our study has some limitations. First, it was a single-centered study, and all patients were frail elderly patients hospitalized for COVID-19, who might be unrepresentative of the general population of older adults, therefore disability and mortality may be over-estimated. Second, the observational design was less robust than an interventional design and prevents any conclusion on causality. Third, during the first wave, PCR diagnosis was not always available and some diagnoses in our study were made using chest CT-scan [31]. However, and despite

relatively low specificity, CT-scan revealed rather good sensitivity to COVID-19 [32, 33]. Fourth, the GIR score is used mainly in France, which does not allow reproducibility of results in other countries, and other more common disability scales should be used in the future to confirm our present results. We could also have taken into account other variables that are now recognised as having an impact on mortality, such as length of hospital stay[34], but which it was not possible to collect in our sample.

## Conclusions

In conclusion, we found that, irrespective of all studied potential confounders, initial functional disability during a SARS-CoV-2 infection was associated with lower 1-year survival among hospitalized geriatric patients. The early assessment of functional disability in older patient with COVID-19 therefore appears crucial to better define the management and benefit/risk ratio of treatments in this population. With the emergence of less severe variants and the availability of vaccines in the frail elderly population, it would be interesting to check whether this result and the prognostic usefulness of disability remains effective in older adults.

## Acknowledgments

The authors wish to thank the GERIA-COVID study group. GERIA-COVID study group: Cédric Annweiler[1], Marine Asfar[1], Melinda Beaudenon[1], Jean Barré[1], Antoine Brangier[1], Mathieu Corvaisier[1], Guillaume Duval[1], Jennifer Gautier[1], Mialy Guenet[1], Jocelyne Loison[1], Frédéric Noublanche[1], Marie Otekpo[1], Hélène Rivière[1], Guillaume Sacco[1], Romain Simon[1].

Affiliations: 1: Department of Geriatric Medicine, University Hospital, Angers, France.

The authors have listed everyone who contributed significantly to the work in the Acknowledgments section. Permission has been obtained from all persons named in the Acknowledgments section.

## Author Contributions

**Conceptualization:** Olivier Brière, Cédric Annweiler.

**Formal analysis:** Olivier Brière, Jennifer Gautier, Cédric Annweiler.

**Investigation:** Olivier Brière, Jennifer Gautier, Cédric Annweiler.

**Methodology:** Olivier Brière, Cédric Annweiler.

**Project administration:** Olivier Brière, Cédric Annweiler.

**Supervision:** Cédric Annweiler.

**Validation:** Olivier Brière, Cédric Annweiler.

**Visualization:** Olivier Brière.

**Writing – original draft:** Olivier Brière, Cédric Annweiler.

**Writing – review & editing:** Olivier Brière, Marie Otekpo, Marine Asfar, Guillaume Sacco, Cédric Annweiler.

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
