## [Decision Letter · Decision Letter 0]

8 Dec 2022

PONE-D-22-32169Initial functional disability as a 1-year prognostic factor in geriatric patients hospitalized for SARS-CoV-2 infectionPLOS ONE

Dear Dr. Briere,

Thank you for submitting your manuscript to PLOS ONE. After careful consideration, we feel that it has merit but does not fully meet PLOS ONE’s publication criteria as it currently stands. Therefore, we invite you to submit a revised version of the manuscript that addresses the points raised during the review process.

We look forward to receiving your revised manuscript.

Kind regards,

Benjamin M. Liu, MBBS, PhD, D(ABMM), MB(ASCP)

Academic Editor

PLOS ONE

Journal Requirements:

Reviewers' comments:

Reviewer's Responses to Questions

**Comments to the Author**

1. Is the manuscript technically sound, and do the data support the conclusions?

Reviewer #1: Partly

Reviewer #2: Yes

2. Has the statistical analysis been performed appropriately and rigorously? 

Reviewer #1: Yes

Reviewer #2: Yes

3. Have the authors made all data underlying the findings in their manuscript fully available?

Reviewer #1: Yes

Reviewer #2: Yes

4. Is the manuscript presented in an intelligible fashion and written in standard English?

Reviewer #1: No

Reviewer #2: Yes

5. Review Comments to the Author

Reviewer #1: 1. Please improve language.

2. Line 43-44: reconstruct the sentences to have clear meaning of background.

3. Line 63: Specify the date of registration for NCT04560608.

4. Line 73-76: make two sentences with clear meaning.

5. It will be nice to strengthen the introduction part by adding some more text.

6. Line 103: the sample size reported is based on what?

7. The participants are aged more than 75 years, are they capable to do the activities mentioned at line 107-111? What about their natural ability? How normal aged and participants previously infected with COVID (same age) are different?

8. Table 1: give the number for male gender. Revise and add data for male participants, this will increase the understanding about the sex wise outcome measures.

Reviewer #2: 

This manuscript attempted to verify Initial functional capacity which is considered an important prognostic marker in older patients hospitalized for COVID-19. The rationale and study design are suitable and good enough to support the author’s hypothesis. However, some points need to be revised to be further considered for publishing.

Comments

Major Comments

First of all the authors depend on grouping of cases into two groups according to 1 year mortality (No and Yes) which cause misleading and the grouping is not clear

It was better to divide your groups according to GIR (GIR score ≤ 3 OR< 3)

Abstract

1- Line 51: "All-cause mortality was collected" did the authors collect only mortality? Or both morbidity & mortality?

Introduction

1- Line 86 what is AGGIR?

Methods

1- Please deliver an explanation about reasons of chosen old patients. Is that information makes any change or add significance idea from authors point of view to the manuscript? And if adult patient may show the same results or not?

2- Sample size is 97 patients which was small (All patients between March and June) why the authors didn`t extend the duration of the study? please provide an explanation for this

3- Authors should provide an explanation for why they didn`t tried to follow up their cases more than just the telephone follow up to serve their research results?

4- Authors should provide an explanation for why the duration of hospital stay not taken in consideration to serve their research results? and if all patient pass through the ICU at any time during their treatment curriculum?

Results

1- Please deliver an explanation about if there is a relation between COVID19 and malignancies as a comorbidity or malignancies are considered an individual cause of morbidity in those old aged, hospitalized patients?

Discussion

1- Please add a brief explanation about the meaning of qSOFA?

2- The authors define disability in their discussion but without providing the readers about most common disabilities the detected during their study, please explain

3- What are the authors recommendations to overcome multiple limitation mentioned in their manuscript?

Minor comments

Abstract

1- Sample size should be mentioned in the method section

Introduction

1- Line 81: reference missed

Methods

1- Please provide subtitles

2- Are there any excluded cases? Reasons and number of excluded cases should be mentioned in detail in the method

Results

Line 153: 88.0+5.4 years, please correct

Line 165: n=96 please correct

Discussion

1- Line 216: [19] italic, please correct

2- Line 242: reference missed

References

Reference 16: begin with" for the VIP2 study group" what did the authors mean?

Finally, the manuscript parts, Introduction, Methods, Results, Tables, and all illustration are done in very good way. I think once authors fix the above comments. The work will be ready to be published in your esteemed journal.

6. PLOS authors have the option to publish the peer review history of their article (what does this mean?). If published, this will include your full peer review and any attached files.

Reviewer #1: **Yes: **Dr. Jayesh J. Ahire

Reviewer #2: No

---

## [Author Response · Author response to Decision Letter 0]

6 Jun 2023

We thank the Editor and Reviewers for the review. As suggested, we have performed a major revision of our manuscript. All the changes are highlighted in yellow on the revised manuscript. Please, find below our point-by-point responses to the comments offered by the Editor and Reviewers. 

Comments of Editor :

Comment 1: “Please ensure that your manuscript meets PLOS ONE's style requirements, including those for file naming. The PLOS ONE style templates can be found at 

https://journals.plos.org/plosone/s/file?id=ba62/PLOSOne_formatting_sample_title_authors_affiliations.pdf”

We have checked and respected your guidelines

Comment 2: “Please provide additional details regarding participant consent. In the ethics statement in the Methods and online submission information, please ensure that you have specified (1) whether consent was informed and (2) what type you obtained (for instance, written or verbal, and if verbal, how it was documented and witnessed). If your study included minors, state whether you obtained consent from parents or guardians. If the need for consent was waived by the ethics committee, please include this information If you are reporting a retrospective study of medical records or archived samples, please ensure that you have discussed whether all data were fully anonymized before you accessed them and/or whether the IRB or ethics committee waived the requirement for informed consent. If patients provided informed written consent to have data from their medical records used in research, please include this information.”

All baseline characteristics were retrospectively collected from patients’ medical records. None of the patients had any objections to the use of anonymised clinical and biological data for research purposes. This is the no-objection principle used in observational studies in France. This information is in the Method section. Please, see on page 5.

Comment 3: “In your Data Availability statement, you have not specified where the minimal data set underlying the results described in your manuscript can be found. PLOS defines a study's minimal data set as the underlying data used to reach the conclusions drawn in the manuscript and any additional data required to replicate the reported study findings in their entirety. All PLOS journals require that the minimal data set be made fully available. For more information about our data policy, please see http://journals.plos.org/plosone/s/data-availability.”

You're absolutely right, it was a mistake on our part. This information was added in the Acknowledgments section. Please, see correction on page 13.

Comments of Reviewer #1:

Comment 1: “Please improve language. Line 43-44: reconstruct the sentences to have clear meaning of background. Line 63: Specify the date of registration for NCT04560608. Line 73-76: make two sentences with clear meaning. It will be nice to strengthen the introduction part by adding some more text.”

The abstract and introduction have been reworded in line with your comment, see correction on page 4.

Comment 2: " Line 103: the sample size reported is based on what?".

The sample size in the study corresponds to the number of patients admitted to the acute geriatric unit at Angers University Hospital, France, during the first wave of the COVID-19 pandemic. All consecutive patients admitted for COVID-19 in the unit were included.

Comment 3: " The participants are aged more than 75 years, are they capable to do the activities mentioned at line 107-111? What about their natural ability? How normal aged and participants previously infected with COVID (same age) are different?”

Not all of the individuals included were able to perform these activities, so they had functional disabilities. They were graded by the AGGIR grid (thus the GIR score) to allow comparison of the prognosis of a SARS-CoV2 infection according to the initial level of disability. 

The study was performed only on geriatric patients in accordance with its main objective. There are no studies to our knowledge that have compared loss of functional abilities in patients < 75 years of age before and after SARS-CoV-2 infection. 

Comment 4: “Table 1: give the number for male gender. Revise and add data for male participants, this will increase the understanding about the sex wise outcome measures.”

Thank you for this comment. It is not the gender, but the sex that is indicated here. We reported the number and proportion of women, which allows a mirror calculation of the number and proportion of men among the participants.

Comments of Reviewer #2:

Comment 1: “First of all the authors depend on grouping of cases into two groups according to 1 year mortality (No and Yes) which cause misleading and the grouping is not clear; It was better to divide your groups according to GIR (GIR score ≤ 3 OR< 3)”

Table 1 has been replaced by splitting the groups according to their functional disability status based on the GIR score. 

Comment 2: Line 51: "All-cause mortality was collected" did the authors collect only mortality? Or both morbidity & mortality?

For our primary endpoint, we focused exclusively on all-cause 1-year mortality.

Comment 3: “Line 86 what is AGGIR?”

The AGGIR grid is commonly used in France in the medical and medico-social sectors to assess dependency in the elderly. In particular, it is used to determine eligibility for state financial support. It is considered robust and reproducible, with no significant inter-operator variability if the assessors have been trained as in our study.

Comment 4: “Please deliver an explanation about reasons of chosen old patients. Is that information makes any change or add significance idea from authors point of view to the manuscript? And if adult patient may show the same results or not?”

The study concerned patients admitted to geriatric wards. It should be noted that SARS-CoV2 infection during the first phase was more frequently symptomatic or severe in the elderly than in younger patients. The majority of hospital admissions therefore concerned elderly people.

It is difficult to imagine the same type of study being carried out on younger patients, as functional disability is much less common.

Comment 5: “Sample size is 97 patients which was small (All patients between March and June) why the authors didn`t extend the duration of the study? please provide an explanation for this”

The sample involved only patients admitted to the geriatrics department of Angers University Hospital, France, and was therefore a monocentric study. In the context of the health crisis, we had to adapt our research methods quickly to respond to urgent issues, and it was difficult to bring together several centers quickly to increase the sample size. 

As the virus had mutated, and the clinical forms had also changed, the results would probably have been different if the patients had been outside the first pandemic phase. 

It would be interesting to repeat the study now, to check whether the results established in this study would still be valid today.

Comment 6: “Authors should provide an explanation for why they didn`t tried to follow up their cases more than just the telephone follow up to serve their research results?”

During the first pandemic phase, we had a lockdown in France, all physical consultations were cancelled, and only urgent hospitalization could be performed. 

It was not possible to monitor patients other than by telephone. If the patient did not reply, we contacted their relatives, and consulted the INSEE national database of all deaths.

Comment 7: “Authors should provide an explanation for why the duration of hospital stay not taken in consideration to serve their research results? and if all patient pass through the ICU at any time during their treatment curriculum?”

We did not collect these two data from all patients during the study, and were therefore not able to include them in the analysis. 

In the study conducted by Leidi et al, it was shown that during the first two phases of the pandemic, the length of hospitalization was linked to hospital mortality [1]. We have added it to the limits of the study. Please, see on page 10.

1. Leidi F, Boari GEM, Scarano O, Mangili B, Gorla G, Corbani A, et al. Comparison of the characteristics, morbidity and mortality of COVID-19 between first and second/third wave in a hospital setting in Lombardy: a retrospective cohort study. Intern Emerg Med. 2022;17: 1941–1949. doi:10.1007/s11739-022-03034-5

Comment 8: “Please deliver an explanation about if there is a relation between COVID19 and malignancies as a comorbidity or malignancies are considered an individual cause of morbidity in those old aged, hospitalized patients?”

In the analysis adjusted for possible confounding factors (Table 2), there is indeed a correlation between the impact of functional disability on survival by adjusting for neoplastic history. They may decrease functional reserve in the elderly hospitalized for COVID-19, and therefore decrease the ability to cope with infection.

Comment 9: “Please add a brief explanation about the meaning of qSOFA?”

This notion was added in the discussion. Please, see on page 9.

Comment 10: “The authors define disability in their discussion but without providing the readers about most common disabilities the detected during their study, please explain”

We didn't have the detailed GIR scores to be able to assess each functional disability separately. This is why we chose to focus on the overall functional disability score.

Comment 11: “What are the authors recommendations to overcome multiple limitation mentioned in their manuscript?”

The study was carried out in the context of a health crisis. It would have been interesting to carry out the study in a larger number of centers, over a longer period, in order to compare the impact of functional disability on survival according to the evolution of the virus.

Comment 12: “Sample size should be mentioned in the method section”

This information was already present in the previous version of the manuscript: “Ninety-seven patients were diagnosed with COVID-19 during the study period in the unit, and were recruited in the GERIA-COVID study. This analysis included all of them.”

Comment 13: “Line 81: reference missed”

There is no reference mentioning the absence of a study on this topic, but in March 2020 there was no study on the topic referenced on PubMed when we designed this study.

Comment 14: “Please provide subtitles”

Thank you for your comment. We made all changes. Please, see on pages 5,6 and 7.

Comment 15: “Are there any excluded cases? Reasons and number of excluded cases should be mentioned in detail in the method”

No patient was excluded from the study.

Comment 16: “Line 153: 88.0+5.4 years, please correct”

Thank you for your comment. The correction was made.

Comment 17: “Line 165: n=96 please correct”

Thank you for your comment. The correction was made.

Comment 18: “Line 216: [19] italic, please correct”

Thank you for your comment. The correction was made.

Comment 19: “Reference 16: begin with" for the VIP2 study group" what did the authors mean?”

This is a group of European researchers working on the management of elderly patients in intensive care. They are interested in the prevalence of major geriatric syndromes: cognition, frailty, dependence and co-morbidities in elderly intensive care patients and their influence on survival. They included 3920 patients in 242 centers across 20 European countries.

---

## [Decision Letter · Decision Letter 1]

17 Jul 2023

Initial functional disability as a 1-year prognostic factor in geriatric patients hospitalized with SARS-CoV-2 infection

PONE-D-22-32169R1

Dear Dr. Briere,

We’re pleased to inform you that your manuscript has been judged scientifically suitable for publication and will be formally accepted for publication once it meets all outstanding technical requirements.

Kind regards,

Benjamin M. Liu, MBBS, PhD, D(ABMM), MB(ASCP)

Academic Editor

PLOS ONE

Additional Editor Comments (optional):

Reviewers' comments:

Reviewer's Responses to Questions

**Comments to the Author**

1. If the authors have adequately addressed your comments raised in a previous round of review and you feel that this manuscript is now acceptable for publication, you may indicate that here to bypass the “Comments to the Author” section, enter your conflict of interest statement in the “Confidential to Editor” section, and submit your "Accept" recommendation.

Reviewer #1: All comments have been addressed

Reviewer #2: All comments have been addressed

2. Is the manuscript technically sound, and do the data support the conclusions?

Reviewer #1: Yes

Reviewer #2: Yes

3. Has the statistical analysis been performed appropriately and rigorously? 

Reviewer #1: Yes

Reviewer #2: Yes

4. Have the authors made all data underlying the findings in their manuscript fully available?

Reviewer #1: Yes

Reviewer #2: Yes

5. Is the manuscript presented in an intelligible fashion and written in standard English?

Reviewer #1: Yes

Reviewer #2: Yes

6. Review Comments to the Author

Reviewer #1: (No Response)

Reviewer #2: The manuscript became suitable and fit well your esteemed journal

Decision Accepted with no further comments

7. PLOS authors have the option to publish the peer review history of their article (what does this mean?). If published, this will include your full peer review and any attached files.

Reviewer #1: **Yes: **Dr. Jayesh J. Ahire

Reviewer #2: **Yes: **Eman A.A. Abdallah

---

## [Editor Report · Acceptance letter]

20 Jul 2023

PONE-D-22-32169R1 

Initial functional disability as a 1-year prognostic factor in geriatric patients hospitalized with SARS-CoV-2 infection 

Dear Dr. Brière:

I'm pleased to inform you that your manuscript has been deemed suitable for publication in PLOS ONE. Congratulations! Your manuscript is now with our production department. 

Kind regards, 

on behalf of

Dr. Benjamin M. Liu 

Academic Editor

PLOS ONE